# Phototoxic Reactions Inducted by Hydrochlorothiazide and Furosemide in Normal Skin Cells—In Vitro Studies on Melanocytes and Fibroblasts

**DOI:** 10.3390/ijms25031432

**Published:** 2024-01-24

**Authors:** Marta Karkoszka, Jakub Rok, Zuzanna Rzepka, Klaudia Banach, Justyna Kowalska, Dorota Wrześniok

**Affiliations:** Department of Pharmaceutical Chemistry, Faculty of Pharmaceutical Sciences in Sosnowiec, Medical University of Silesia, Jagiellońska 4, 41-200 Sosnowiec, Poland; marta.karkoszka@sum.edu.pl (M.K.); jrok@sum.edu.pl (J.R.); zrzepka@sum.edu.pl (Z.R.); kbanach@sum.edu.pl (K.B.); jkowalska@sum.edu.pl (J.K.)

**Keywords:** hydrochlorothiazide, furosemide, melanocytes, fibroblasts, phototoxicity

## Abstract

Hypertension is known to be a multifactorial disease associated with abnormalities in neuroendocrine, metabolic, and hemodynamic systems. Poorly controlled hypertension causes more than one in eight premature deaths worldwide. Hydrochlorothiazide (HCT) and furosemide (FUR), being first-line drugs in the treatment of hypertension, are among others the most frequently prescribed drugs in the world. Currently, many pharmacoepidemiological data associate the use of these diuretics with an increased risk of adverse phototoxic reactions that may induce the development of melanoma and non-melanoma skin cancers. In this study, the cytotoxic and phototoxic potential of HCT and FUR against skin cells varied by melanin pigment content was assessed for the first time. The results showed that both drugs reduced the number of metabolically active normal skin cells in a dose-dependent manner. UVA irradiation significantly increased the cytotoxicity of HCT towards fibroblasts by approximately 40% and melanocytes by almost 20% compared to unirradiated cells. In the case of skin cells exposed to FUR and UVA radiation, an increase in cytotoxicity by approximately 30% for fibroblasts and 10% for melanocytes was observed. Simultaneous exposure of melanocytes and fibroblasts to HCT or FUR and UVAR caused a decrease in cell viability, and number, which was confirmed by microscopic assessment of morphology. The phototoxic effect of HCT and FUR was associated with the disturbance of redox homeostasis confirming the oxidative stress as a mechanism of phototoxic reaction. UVA-irradiated drugs increased the generation of ROS by 10–150%, and oxidized intracellular thiols. A reduction in mitochondrial potential of almost 80% in melanocytes exposed to HCT and UVAR and 60% in fibroblasts was found due to oxidative stress occurrence. In addition, HCT and FUR have been shown to disrupt the cell cycle of normal skin cells. Finally, it can be concluded that HCT is the drug with a stronger phototoxic effect, and fibroblasts turn out to be more sensitive cells to the phototoxic effect of tested drugs.

## 1. Introduction

Adverse drug reactions (ADRs) are known to be the top 10 causes of disease and death in developed nations among patients undergoing occasional and chronic pharmacotherapy. Moreover, ADRs contribute to increased healthcare costs by ca. 9%, hospitalization admissions by about 6% as well as drug retractions from the pharmaceutical market, becoming a serious health problem worldwide [1]. Among others, clinical manifestations of ADRs are often accompanied by skin side effects [2]. The two main types of drug-induced skin hypersensitivity are photoallergy and phototoxicity [3]. Photosensitivity is known to be the third most common ADR [4]. Even though the precise incidence of photo-hypersensitivity reactions is not accurately defined due to diagnostic difficulties, about 5–16% of referrals to dermatology centers are caused by phototoxic adverse reactions, while ca. 2–8% is the result of photoallergy [5]. A cutaneous phototoxic reaction is the result of the use of a drug with photosensitizing potential and subsequent exposure to UV irradiation [6,7]. From a clinical point of view, phototoxicity usually resembles sunburn, and symptoms of this ADR include edema, tenderness, erythema, blistering, and burning. The most important features describing phototoxic reactions are dose-dependent effect, occurrence shortly after sun exposition, and localization on the UV-exposed areas [8].

Currently, about 300 medications are known to have the ability to induce phototoxic reactions. The most important aspect determining the phototoxic potential of a molecule is the ability of the basic compound or its metabolite to absorb irradiation. Common features of many widely known phototoxic drugs are the presence of unstable double bonds, halogen substituents, heteroatoms, or tricyclic configuration in the chemical structure allowing for reaching an excited state [6]. Because the excited state of the molecule is characterized by high energy, it is short-lived, and the return to the ground state is associated with the generation of reactive oxygen species, causing direct damage to cellular and tissue structures through various mechanisms [9]. Additionally, ineffective neutralization of ROS by intracellular enzymatic and non-enzymatic antioxidants can lead to proteins, DNA, and lipids oxidative damage, as well as dysregulations of intracellular metabolic activity [10]. Nowadays, among the wide list of drugs characterized by high phototoxic potential containing NSAIDs, tetracyclines, or fluoroquinolones, there are also drugs with a sulfonamide moiety in chemical structure, such as thiazide diuretics, e.g., hydrochlorothiazide, and furosemide which are the subject of this study [11,12].

Hydrochlorothiazide and furosemide have been commonly prescribed medications in the treatment of hypertension since the mid-twentieth century in both Europe and the USA [13]. The first reports of side effects related to phototoxicity appeared shortly after their introduction into public treatment [14]. Actually, there are many emerging reports, cohort, and retrospective studies linking the use of diuretics, especially hydrochlorothiazide, with an increased risk of developing skin cancers. It is suggested that the described phenomenon is directly related to the promotional properties of HCT to increase the absorption of UV radiation, which is the main risk factor for all skin cancers [3,15,16].

The diversity of skin cells in terms of the content of melanin biopolymers predisposes them to different responses to the phototoxic effects of drugs. The perinuclear location of melanin and its ability to absorb and disperse UV radiation means that the primary function of this pigment is to protect the genetic material against harmful exogenous factors [8]. The lower content of melanin pigments in the skin predisposes Caucasians to greater susceptibility to phototoxic reactions and the development of skin cancer [15]. However, the possibility of forming drug-melanin complexes leads to the accumulation of the drug in the skin and the occurrence of much higher concentrations of drug substances than in the plasma, directly affecting the safety and effectiveness of used pharmacotherapy [17].

Although it is proven that low doses of diuretics reduce mortality and cardiovascular morbidity, the effect of chronic pharmacotherapy on the homeostasis of human normal skin cells has not been investigated so far. In addition, the cellular and molecular basis of skin tumor development are still unknown, but it is supposed that the increase in carcinogenesis may be related to hydrochlorothiazide-induced phototoxic effects [18].

Taking into account the above facts, the study aimed to assess the effect of two diuretics—HCT, and FUR on homeostasis and redox balance of human normal skin cells with different content of melanin biopolymers.

## 2. Results

### 2.1. The Phototoxic Effect of Hydrochlorothiazide or Furosemide Decrease the Number of Metabolic Activity and Induce Morphological Changes of Normal Skin Cells

A preliminary analysis of phototoxicity of tested diuretics was performed with WST-1 reagent. Before the irradiation, cells were incubated with HCT or FUR in wide range of concentrations 0.01 mM–1.0 mM for 24 h. Metabolic activity of melanocytes and fibroblasts were assessed 24 h after UVAR exposure. The obtained results, depicted in Figure 1, showed that both diuretics decreased the number of metabolically active cells proportionally to the used concentration. The simultaneous combination of the drug and UVAR significantly enhanced the obtained effect. It was noticed that HCT and FUR showed similar cytotoxic potential.

A significant decrease in the number of metabolically active cells was observed after the treatment of unirradiated cultures with HCT—approximately 20% decrease was noted in the case of fibroblasts and melanocytes. Moreover, the obtained values suggested that fibroblasts were more sensitive to the phototoxic effect of the tested diuretics than melanocytes. The strongest phototoxic effect (ca. 65% decrease compared to the control) was observed after the application of 1.0 mM HCT and UVAR on fibroblasts. Comparing, FUR in the highest concentration caused an approx. 35% decrease in the number of metabolically active fibroblasts, and ca. 20% decrease in the case of melanocytes.

The results were reflected in microscopic images of the cell cultures (Figure 1E). The depicted photographs showed morphological changes that occurred as a result of cell exposure to the drug and/or irradiation. In all cultures exposed to the drugs and UVAR, a decrease in cell number, loss of intercellular contacts and changes in cell shape were observed compared to controls.

Considering the obtained results of the WST-1 analysis, further experiments were performed using drug concentrations of 0.5 mM and 1.0 mM.

### 2.2. The Influence of Hydrochlorothiazide or Furosemide and/or UVA Irradiation on Cell Number and Viability

In the next stage of the study, the effect of both diuretics on viability (Table 1) and cell number (Figure 2) was assessed using the imaging cytometry. Based on the results, it was found that the greatest decrease in the percentage of viable cell was observed in fibroblasts exposed to HCT and UVAR simultaneously. In this sample, an approximately 41% decrease in cells viability was observed. The exposure of melanocytes to FUR and UVAR decreased cell viability by approximately 11% compared to the cells treated only with FUR. Moreover, the data obtained showed that this drug did not affect the viability of fibroblasts.

In all samples treated with the drug at a concentration of 1.0 mM and irradiated, an approximately 50% decrease in cell number was observed (Figure 2). The UVA irradiation contributes to the decrease in number of melanocytes by ca. 35% compared to cells treated with HCT in the concentration of 1.0 mM. The greatest effect of reducing of the cell number was observed in cultures of fibroblasts exposed to FUR and UVAR simultaneously—about 58%, and the smallest in the case of melanocytes exposed to FUR with UVAR—about 35%. Separate application of UVA radiation or drugs did not affect the number of normal skin cells. It was observed that HCT decreased the number of fibroblasts in a dose-dependent manner.

### 2.3. The Assessment of Changes in Transmembrane Mitochondrial Potential (TMP) in Melanocytes and Fibroblasts Exposed to UVA Irradiation and Hydrochlorothiazide or Furosemide

The JC-1 dye was used to assess TMP in normal skin cells treated with HCT or FUR and/or exposed to UVA radiation. The analysis presented in Figure 3 showed that HCT induced depolarization of mitochondrial membranes of fibroblasts and melanocytes in a concentration-dependent manner, causing an increase in the percentage of cells with a depolarized mitochondrial membrane as the concentration increases. UVA radiation is a factor that enhances the observed effect. Fibroblasts turned out to be more sensitive to the effects of HCT and UVAR than melanocytes, as the percentage of cells with low TMP was about 80% for fibroblasts, and about 60% in the case of pigmented cells. Exposure of skin cells to furosemide and UVAR caused significant changes in TMP only in fibroblasts. An approx. 35% increase in the percentage of fibroblasts with low TMP was observed in contrast to melanocytes where the difference was only 5%.

### 2.4. The Evaluation of the Changes in the Cell Cycle of Melanocytes and Fibroblasts Exposed to Furosemide or Hydrochlorothiazide and/or UVAR

Due to the fact that one of the possible reasons for the inhibition of the metabolic activity of cells exposed to the drug and/or UVAR may be cell cycle disorders, a cytometric analysis of this parameter was performed. The cell cycle analysis of normal skin cells was carried out after 24-h exposure of cell cultures to UVAR by measuring the fluorescence intensity emitted by DAPI specifically bound to the genetic material of the tested cells. The obtained data were visualized as representative histograms and bar graphs presenting the percentage of cells in the various phases of the cell cycle (Figure 4).

The obtained results allow us to conclude that UVA irradiation alone changes the cell cycle profile of both fibroblasts and melanocytes. In cell cultures exposed only to UVAR, a significant reduction in the percentage of cells in the G1/G0 phase and an increase in the S and G2/M phases were noted. In the case of fibroblasts treated with HCT or FUR, it was observed that exposure to the drug contributed to the enhancement of the observed changes. In addition, it was found that the exposure of cells to the drug at a dose of 1.0 mM and UVAR caused an increase in the percentage of cells in the subG1 phase, which indicated DNA fragmentation. The highest percentage of cells in the subG1 phase, about 16%, was observed in fibroblasts treated with HCT and exposed to UVAR. For melanocytes, the main observed change for a drug and/or UVAR, as well as only UVAR exposure is a decrease in the percentage of cells in the G1/G0 phase. The slight increase in the subG1 phase to approximately 7% was found only in irradiated cells treated with HCT or FUR.

### 2.5. The Quantitative Analysis of the Effect of Hydrochlorothiazide or Furosemide and/or UVA Irradiation on Redox Homeostasis of Normal Skin Cells

Intracellular oxidative stress is a result of an imbalance between the generated reactive oxygen species (ROS) and the enzymatic and non-enzymatic capacity of the cell for their neutralization. To determine the effect of HCT or FUR and/or UVA radiation on the occurrence of oxidative stress in fibroblasts and melanocytes, the H_2_DCFDA test was performed (Figure 5). The method is based on the quantitative measurement of the ROS level by staining the cells with a reduced form of fluorescein, which in an oxidized state exhibits green fluorescence. The conducted analysis showed that both FUR and HCT caused an approximately 50% increase in ROS in melanocytes and fibroblasts. UVA irradiation contributes to the intensification of the obtained effect, increasing ROS production, proportional to the drug concentration. However, the effect was not observed in irradiated melanocytes treated with FUR. The highest increase (by about 150%) in the level of ROS was in fibroblasts exposed to HCT and UVR. Similarly, in irradiated melanocytes cultured with HCT, the ROS level increased by approximately 100%. H_2_DCFDA analysis showed that FUR in combination with UVAR caused less disturbance of redox homeostasis of skin cells than HCT. In the case of melanocytes, changes in the level of ROS were observed only at the highest concentration and UVAR used, demonstrating the ability of the cells to defend against the phototoxic effect of the drug.

### 2.6. The Effect of Hydrochlorothiazide or Furosemide and/or UVA Irradiation on Intracellular Reduced Thiols Status in Melanocytes and Fibroblasts

Glutathione (GSH), one of the most important intracellular thiol compounds, occurs in a reduced form in conditions of oxidoreductive homeostasis, indicating the high level of cell vitality. To assess the effect of HCT or FUR and/or UVAR on the intracellular level of reduced thiols, a cytometric analysis was performed using a specific thiol-reactive fluorescent dye VitaBright-48™. Based on the obtained data (Figure 6), it can be concluded that tested drugs affect fibroblasts to a greater extent than melanocytes. The percentage of fibroblasts with a low level of reduced thiols was increased to about 30% after the treatment with HCT at a concentration of 1.0 mM. The additional use of UVA radiation in this case rose the value to about 50%. Similarly, FUR at 1.0 mM and UVAR caused an increase of approximately 30% over the control. In the case of melanocytes, an approx. 30% increase in the percentage of cells with low levels of reduced thiols treated with HCT or FUR at a concentration of 1.0 mM and irradiated with UVA was observed. The obtained results indicate that fibroblasts are more sensitive cells to the phototoxic effects of tested drugs. Moreover, the described changes are reflected in the analysis of the ROS level.

## 3. Discussion

Drug-induced photosensitivity, which is a dermatological growing problem, is one of the most common cutaneous ADRs type [8]. The reasons for the increasing occurrence of phototoxicity cases include an increased content of substances with phototoxic potential in food and cosmetic products and more frequent exposure to UV radiation caused by the preference for dark skin tone. Also, knowledge about the common deficiency of vitamin D3 makes society more often expose its skin to UV radiation in order to equalize the its level. In addition, there are many over-the-counter drugs from various therapeutic groups available in the pharmaceutical market, which increase the percentage of society using pharmacotherapy without appropriate medical care. Therefore, occurring cutaneous ADRs contribute to patients discontinuing pharmacotherapy or using additional drugs aimed at alleviating the accompanying ADRs, directly affecting its effectiveness [19,20].

Hypertension is known to be a growing health problem worldwide. It is a complex disease associated with multiple abnormalities in metabolic, hemodynamic, and neuroendocrine systems leading to the development of cardiovascular diseases [21,22]. The first-line drugs recommended for the treatment of hypertension both in United Stated and Europe include diuretics like hydrochlorothiazide (HCT) and furosemide (FUR) [14,21,23,24,25]. Due to the fact that multi-morbidity associated with polypharmacy mainly affects the elderly population with an increased risk of heart disease, cutaneous side effects mainly concern this social group. It is worth noting that effective treatment of hypertension involves the use of several drugs such as beta-blockers, diuretics and calcium channel blockers, most of which have proven phototoxic properties. Exposure of the skin to many factors inducing phototoxic reactions contributes to the skin’s sensitivity to the toxic effects of UV radiation, predisposing it to the occurrence of phototoxic reactions and the induction of damage to the genetic material that may result in the development of cancer [26,27].

Although both commonly known and used drugs such as HCT and FUR are initially subjected to phototoxicity test, many new reports, systematic reviews, and cohort studies of phototoxic reactions have been reported as an adverse effects of pharmacotherapy with these drugs [14,27,28,29]. Moreover, a lot of recent pharmacoepidemiological data have appeared to link a dose-dependent, cumulative association between long-term HCT treatment and increased risk of skin cancer [24,27,28,29]. The meta-analyses of the relationship between pharmacotherapy with thiazide diuretics and the risk of skin cancer demonstrated an increased risk of developing non-melanoma skin cancers—squamous cell carcinoma (SCC) and basal cell carcinoma (BCC), for which the OR value was 1.86 and 1.19, respectively, and melanoma cancer (OR 1.14) [30]. This study confirmed the results of the presented meta-analyses at the molecular level by demonstrating greater resistance of melanocytes to the phototoxic effect of diuretics compared to fibroblasts. Based on the available WHO data, special precautions and warnings for the use of medications containing HCT have been added to the Summary of Product Characteristics in accordance with the European Medicine Agency Pharmacovigilance Risk Assessment Committee recommendations [24,31,32,33,34].

Despite the fact that HCT and FUR are drugs used mainly in the treatment of hypertension, renal dysfunction and edema, their phototoxic ADR is widely known [35]. However, currently available data do not indicate a direct effect of these drugs on normal skin cells. In this study, the cytotoxic and phototoxic potential of HCT and FUR against human normal skin cells with different content of melanin pigments was analysed for the first time. It was found that simultaneous exposure of human normal skin cells causes disturbances of intracellular redox homeostasis which was confirmed by quantifying the ROS levels and the assessment of reduced intracellular thiols. Furthermore, differences in cell cycle and mitochondrial potential induced by irradiated HCT and FUR in normal skin cells with different degrees of pigmentation were assessed. On the one hand, the photoprotective properties of melanins are related to their perinuclear localization, aimed at protecting DNA, but on the other hand, melanin has antioxidant properties, protecting the cell against the harmful effects of ROS [17,36,37]. Moreover, melanin biopolymers are able to bind and retain in the skin many substances, including drugs, creating long-term reservoir [38]. The formation of drug-melanin complexes may result in much higher concentrations of the drug in the skin, affecting the effectiveness and safety of the treatment, and increasing probability of phototoxic reactions occurrence. Drugs with confirmed ability to bind to melanins are NSAIDs, tetracyclines, and fluoroquinolones [39,40,41]. The observed changes demonstrated in this study suggest a significant contribution of melanin to the cellular response to HCT, FUR, and UVAR. It seems that melanin has photoprotective properties against the tested drugs, as fibroblasts turned out to be more sensitive cells to the tested agents, which is confirmed by the results of the studies discussed below and other tests conducted in vivo [42,43,44]. A similar experimental model was previously used to investigate the phototoxicity of fluoroquinolones and meloxicam against human normal skin cells. It has been shown that the tested drugs used in combination with UVA inhibit the proliferation of fibroblasts and melanocytes, and the obtained effect depends on the concentration of the analysed substances [45,46]. The presented results showing phototoxic properties of HCT and FUR was obtained during various analysis, including the number and viability of cells as well as microscopic images.

It is worth noting that other studies have proven that drugs containing a sulphonamide moiety, such as hydrochlorothiazide, cause skin reactions during the simultaneous irradiation with UVA. The changes within the skin induced by the phototoxicity are visible both macroscopically and histologically. Fibroblast morphology disorders including changes in cells shape, loss of intracellular connections and cell number reduction may indicate a significant participation of this group of cells in skin damage caused by HCT. Moreover, an in vitro assay analyzing the phototoxic potential of HCT indicated that the drug with UVAR caused dysplastic morphological changes in the keratinocytes HaCaT cell line [47].

Based on the obtained data, it was found that HCT, and FUR caused a reduction in the number of melanocytes and fibroblasts. The simultaneous irradiation with UVA contributed to the enhancing the effect. The observed decrease in the cell number may be the result of the inhibition of cell cycle. In all analyzes, the tested drugs reduced the percentage of cells in the G1/G0 phase, and UVA radiation additionally lowered the number of cells in this phase. In addition, the phototoxic action of HCT and FUR led to genetic material damage and DNA fragmentation as evidenced by an increase in the percentage of cells in the subG1 phase. Similar changes in the cell cycle were caused by phototoxic action of meloxicam in melanocytes and fibroblasts—the drug caused a decrease in the percentage of cells in the G1/G0 phase [45]. The analysis of the effect of lomefloxacin and UVAR on the fibroblast cell cycle showed similar results relating to the percentage of cells in the G1/G0 and G2/M phases as the HCT and FUR studies. Conversely, a decrease in the percentage of fibroblasts in the S phase was observed [48]. Although cell cycle disruption is one of the symptoms of phototoxicity, the direction of change may depend on the tested drug and used cell line.

Currently, two basic types of phototoxic reaction are distinguished—type I, oxygen dependent, (photodynamic), and type II, which does not depend on oxygen, (non-photodynamic) [8,49]. The first type of phototoxic reactions is accompanied by the generation of a large amount of ROS, which include hydroxyl peroxide, superoxide anion, as well as hydroxyl radical. The second type, non-photodynamic reaction, manifests in the generation of highly toxic and reactive singlet oxygen [50,51,52]. Both analysed diuretics are known to be photolabile compounds under aerobic and anaerobic conditions. During the irradiation, they are converted to unstable photoproducts, which contributes to the induction of phototoxicity reactions [47,50]. This study proved that HCT and FUR triggered generation of ROS in tested cultures, but their potential was different. HCT itself, in a dose-dependent manner, induced the production of ROS in both analysed cell lines, while FUR exhibited such properties only in relation to fibroblasts. In addition, HCT demonstrated a significantly greater pro-oxidative effect than FUR in irradiated cells. Significant differences in the results of H_2_DCFDA assay were noticed when compared cells treated only with the drug to cells exposed to the drug and UVAR. The results clearly indicated that UVA radiation is a factor enhancing the generation of ROS in treated cells. In addition, it was shown that the intracellular redox homeostasis of normal skin cells was disturbed, as evidenced by the high level of oxidized thiols. An imbalance between the amount of ROS generated and the cell ability to neutralize them is referred to as oxidative stress. The obtained results allow us to conclude that HCT is the drug causing greater oxidation-reduction disorders than furosemide, while fibroblasts are more sensitive cells to oxygen homeostasis disorder than melanocytes. Many drugs with high phototoxic potential, i.e., lomefloxacin or meloxicam, induced disturbances of redox homeostasis in skin cells which resulted in an increase in ROS levels, a decrease in the percentage of cells with high levels of reduced thiols, similar to the tested HCT and FUR [45,48].

Considering the multitude of processes taking place with the participation of ATP-producing organelles, mitochondria are known to be the primary source of intracellular ROS [53,54]. ROS generated by mitochondria may regulate cell proliferation, among others by the induction of cell cycle arrest. Reduced polarization of mitochondrial membranes is observed in unfavorable conditions and is contributed by, among others, to induce oxidative stress and apoptosis [55]. The results presented in this article demonstrated that the tested drugs exhibited different potential on the influence on TMP. HCT turned out to reduce the mitochondrial membrane potential to a much greater extent than FUR. FUR lowered the mitochondrial membrane potential only in fibroblasts during the simultaneous UVA irradiation. The obtained results allow us to conclude that one of the phototoxic effects of analysed diuretics on human normal skin cells is mitochondrial membrane depolarization.

In this study, for the first time, the cyto- and phototoxic potential of hydrochlorothiazide and furosemide was analysed on two human normal skin cell lines—melanocytes and fibroblasts, differentiated in terms of the content of melanin pigments. The obtained results showed that both drugs have antiproliferative potential against melanocytes and fibroblasts, and UVA radiation additionally contributes to its deepening. These results are reflected in the decrease in cell number and proliferation as well as the observed morphological changes. Simultaneous exposure of skin cells to the tested drugs and UVAR resulted in changes in the percentage of cells in individual phases of the cell cycle, a decrease in the transmembrane mitochondrial potential and changes in intracellular redox homeostasis. The tested drugs and UVAR significantly increase the level of intracellular ROS and reduce the percentage of cells with high levels of reduced thiols. The presented results indicate the phototoxic properties of the tested drugs; however, hydrochlorothiazide is a drug with a greater phototoxic potential and fibroblasts are more sensitive cells to the tested factors than melanocytes. We believe that these data provide a strong basis for further research on the phototoxic effects of furosemide and hydrochlorothiazide.

## 4. Materials and Methods

### 4.1. Chemicals and Reagents

Hydrochlorothiazide, amphotericin B, penicillin, Fibroblasts Growth Medium, and phosphate-buffered-saline were obtained from Sigma Aldrich Inc. (Taufkirchen, Germany). Furosemide (20 mg/2 mL) was purchased from Polpharma (Warszawa, Poland). Melanocytes growth medium M-254 as well as human melanocyte growth suppelement-2 (HMGS-2) were acquired from Cascade Biologics (Portland, OR, USA). Neomycin was purchased from Amara (Kraków, Poland). Trypsin/EDTA was acquired from Cytogen (Zgierz, Poland). Via-1-Cassettes™ (acridine orange and DAPI fluorophores), NC-slides A8, Solution 3 (1 µg/mL DAPI, 0.1% triton X-100 in PBS), Solution 5 (VB-48™ PI AO), Solution 7 (200 µg/mL JC-1) and Solution 8 (1 µg/mL DAPI in PBS) were obtained from ChemoMetec (Lillerød, Denmark). H_2_DCFDA reagent was purchased from Thermo Fisher Scientific Inc. (Waltham, MA, USA). WST-1 cell proliferation reagent was obtained from Roche GmbH (Mannheim, Germany). The other remaining chemicals were acquired from Sigma Aldrich Inc. (Taufkirchen, Germany) or POCH S.A. (Gliwice, Poland).

### 4.2. Cell Culture, Treatment and Exposure to UVA Irradiation

All studies were performed on human dermal fibroblasts (HDF) which were acquired from Sigma Aldrich Inc. (St. Louis, MO, USA), and human epidermal melanocytes (HEMn-LP) purchased from Cascade Biologics (Portland, OR, USA). HDF was cultured in a ready-to-use fibroblast growth medium. Melanocytes were maintained in medium M-254 with the addition of HMGS-2 and neomycin (10 μg/mL), amphotericin B (0.25 mg/mL), and penicillin G (10,000 U/mL). Melanocytes and fibroblasts were seeded into Petri dishes (7.5 × 105 cells/dish) and 96-well microplates (5000 cells/well). Cells treatment with HCT and FUR began 48 h after seeding. The cells were exposed to HCT and FUR solutions for 24 h. After replacing the drug solutions or medium with PBS, cells were irradiated with UVA at a dose 5 J/cm^2^ using a lamp BVL-8.LM (VilberLourmat, Collégien, France). Non-irradiated samples were kept in PBS in the dark. Subsequently, PBS was removed and the tested cells were cultured in appropriate growth medium for another 24 h.

### 4.3. The Assessment of Cells’ Metabolic Activity

The metabolic activity of human skin cells was assessed with the WST-1 reagent. The colorimetric analysis is based on the ability of metabolically active cells to convert tetrazolium salt to formazan. Thus, the quantity of formazan dye produced correlates to the amount of metabolically active cells. Briefly, the tested cells were seeded in 96-well microplates in an appropriate growth medium, and then treated with HCT and FUR solutions ranging from 0.01–1.0 mM for 24 h. The UVA irradiation procedure was carried out as described in Section 2.2. Subsequently, 10 µL of the WST-1 solution was added to every well. The sample absorbance was measured after 3 h incubation period at 37 °C and 5% CO_2_ at 440 nm and 650 nm with Microplate reader Infinite 200 PRO (TECAN, Männedorf, Switzerland).

### 4.4. The Analysis of Cell Number and Viability

Image cytometer NucleoCounter^®®^ NC-3000™ (ChemoMetec, Lillerød, Denmark) was used to analyze the number and viability of normal skin cells exposed to HCT or FUR and/or UV radiation type A (UVAR). Briefly, the cell cultures were trypsinized, centrifuged, and subsequently resuspended in the adequate growth medium. Afterward, cell samples were drawn into Via1-CassetteTM which included DAPI (staining the population of non-viable cells) and acridine orange (all cell detection). The analyses were carried out in accordance with Cell Viability and Cell Count Assay.

### 4.5. Mitochondrial Potential Analysis

The mitochondrial potential analysis was performed using the NucleoCounter^®®^ NC-3000™ imaging cytometer according to the procedure described earlier [45,56]. The ability of JC-1 to accumulate in the mitochondrial matrix of cells with high transmembrane mitochondrial potential and the emission of red fluorescence allowed to determine the percentage of normal cells. The accumulation of monomeric forms of JC-1 in the cytoplasm and the green fluorescence are characteristic of cells with a depolarized mitochondrial membrane allowing for the assessment of the percentage of cells with low mitochondrial potential. Solution 7 was added to the obtained samples and incubated for 15 min at 37 °C. Immediately before the analysis 250 µL of Solution 8 was added to the cell pellets and measured using the NucleoView NC-3000 software (ChemoMetec, Denmark).

### 4.6. The Assessment of the Intracellular Content of Reduced Thiols

Intracellular reduced thiol levels in melanocytes and fibroblasts were measured using the image cytometer NucleoCounter^®®^ NC-3000™ following the protocol described previously [45,56]. The analysis is based on the use of VitaBright-48™ dye, which reacts with reduced thiols to create a fluorescent product. The fluorescence intensity depends on the level of reduced thiols in the cells. Cell pellets were suspended in 190 µL of appropriate growth medium and 10 µL of Solution 5 was added. Prepared samples were analysed immediately with a cytometer. The obtained histograms allowed to demarcate the percentage of cells with low (dead cells) and high (healthy cells) intracellular GSH levels.

### 4.7. Cell Cycle Assessment

Imaging cytometry technique was used to analyze the cell cycle of normal skin cells following the previously described procedure [45,56]. Briefly, after the treatment and UVAR procedure, the cells were harvested, counted, and fixed with 70% ethanol at 0–4 °C for 24 h. Subsequently, the ethanol was removed and cell pellets were stained with Solution 3. In cells permeabilized with ethanol, low-molecular-weight DNA is released from apoptotic cells. Subsequently, low-molecular-weight DNA is removed from the sample, and high-molecular-weight DNA retained in the cells is stained with DAPI dye. Fixed Cell Cycle-DAPI Assay protocol was used to analyze prepared samples.

### 4.8. H_2_DCFDA Assay–Reactive Oxygen Species Quantitation

The H_2_DCFDA fluorescent reagent was used to quantify intracellular ROS content. The analysis was carried out in accordance with the procedure described in an earlier article [45]. The assay involves staining cells with 2′,7′-dichlorodihydrofluorescein diacetate (H_2_DCFDA), which is a reduced form of 2′,7′-dichlorofluorescein diacetate (DCFDA). H_2_DCFDA is deacetylated by intracellular esterases to H_2_DCF and then oxidized to 2′,7′-dichlorofluorescein (DCF), which exhibits green fluorescence. Cells after the HCT and FUR treatment and/or UVAR exposure were incubated with the H_2_DCFDA reagent for 30 min without light access. Subsequently, the cells were washed twice with PBS and fluorescence intensity was measured with Infinite 200 Pro microplate reader. The obtained results were expressed as a percentage of control cells.

### 4.9. Statistical Analysis

Statistical analysis was performed using GraphPad Prism 6.01. In all experiments, mean values of at least three separate experiments performed in triplicate (n = 9) ± standard deviation (SD) were calculated. Differences among groups were assessed using two-way ANOVA analysis of variance followed by Dunnett’s test; *p* < 0.05 was determined to indicate a significant difference.

## Figures and Tables

**Figure 1 ijms-25-01432-f001:**
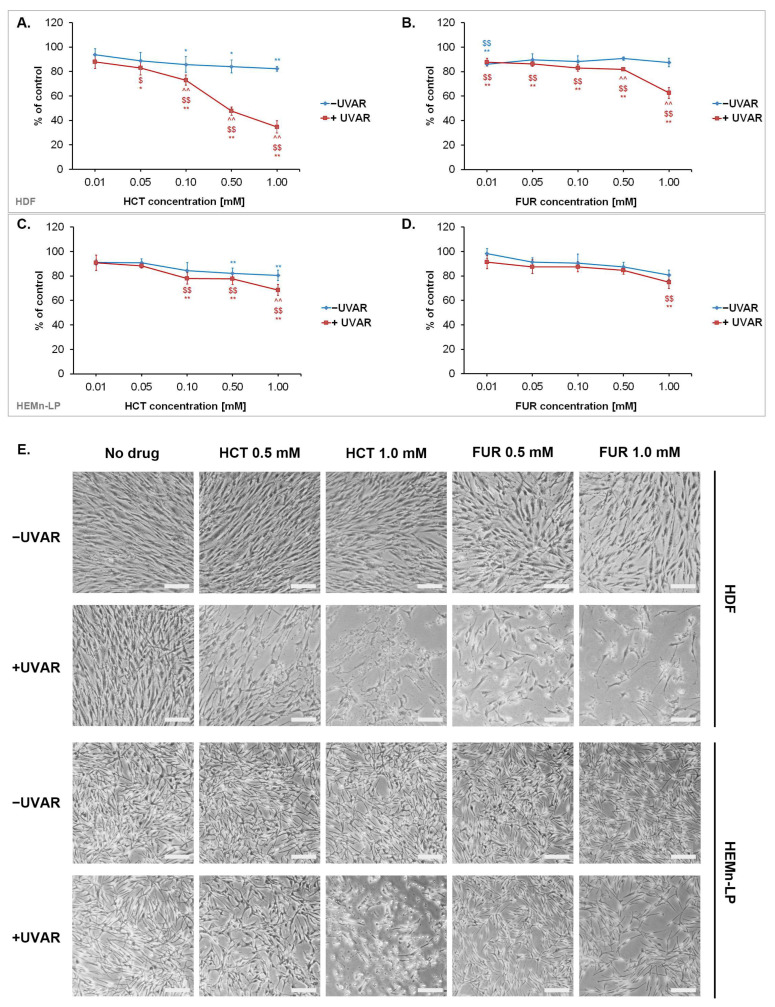
The evaluation of cytotoxic and phototoxic effects of hydrochlorothiazide and furosemide. Metabolic activity of skin cells—fibroblasts (**A**,**B**) and melanocytes (**C**,**D**) was assessed on the basis of the WST-1 assay. Representative photographs showing morphological changes in normal skin cells after the investigated treatment, scale bar = 200 µm (**E**). * *p* < 0.05, ** *p* < 0.01 vs. untreated cells (control); $ *p* < 0.05, $$ *p* < 0.01 vs. UVA-irradiated cells (non-exposed to the drug); ^^ *p* < 0.01 vs. corresponding sample not irradiated with UVAR.

**Figure 2 ijms-25-01432-f002:**
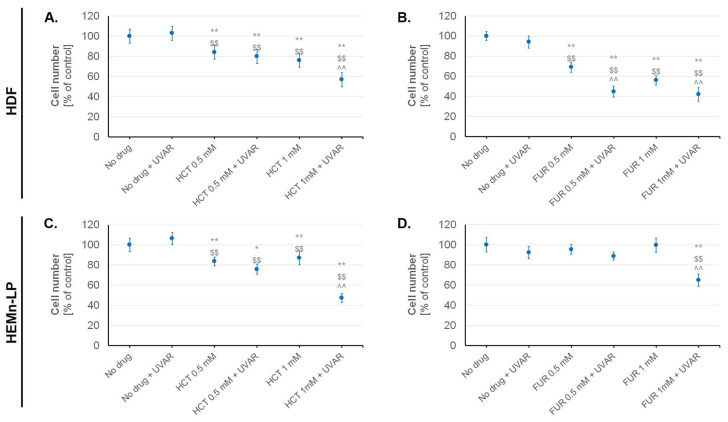
Analysis of the number of normal skin cells—fibroblasts (**A**,**B**) and melanocytes (**C**,**D**) exposed to hydrochlorothiazide and furosemide, as well as UVA irradiation. * *p* < 0.05; ** *p* < 0.01 vs. untreated cells (control); $$ *p* < 0.01 vs. irradiated cells (non-treated cells); ^^ *p* < 0.01 vs. corresponding sample not treated with UVAR.

**Figure 3 ijms-25-01432-f003:**
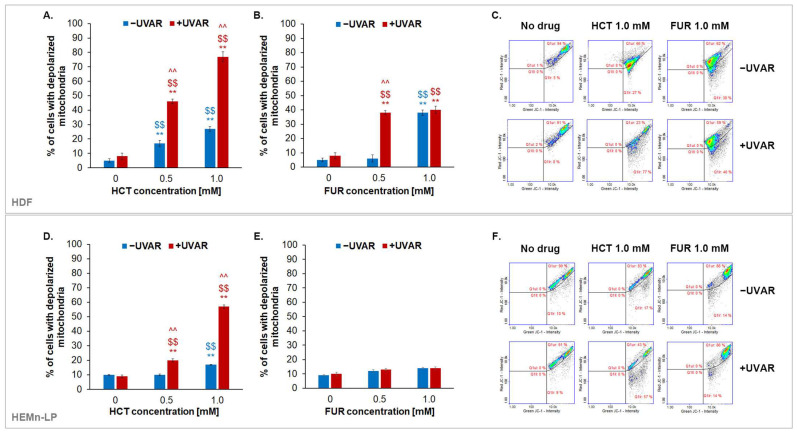
The transmembrane mitochondrial potential of fibroblasts (**A**–**C**) and melanocytes (**D**–**F**) treated with hydrochlorothiazide (0.5 mM and 1.0 mM) or furosemide (0.5 mM and 1.0 mM) and/or exposed to UVAR. Representative scatter plots showing cells differentiated into subpopulations: cells with reduced mitochondrial membrane potential (green), and cells with polarized mitochondrial membrane (red). ** *p* < 0.01 vs. untreated cells (control); $$ *p* < 0.01 vs. irradiated cells (non-treated cells); ^^ *p* < 0.01 vs. corresponding sample not treated with UVAR.

**Figure 4 ijms-25-01432-f004:**
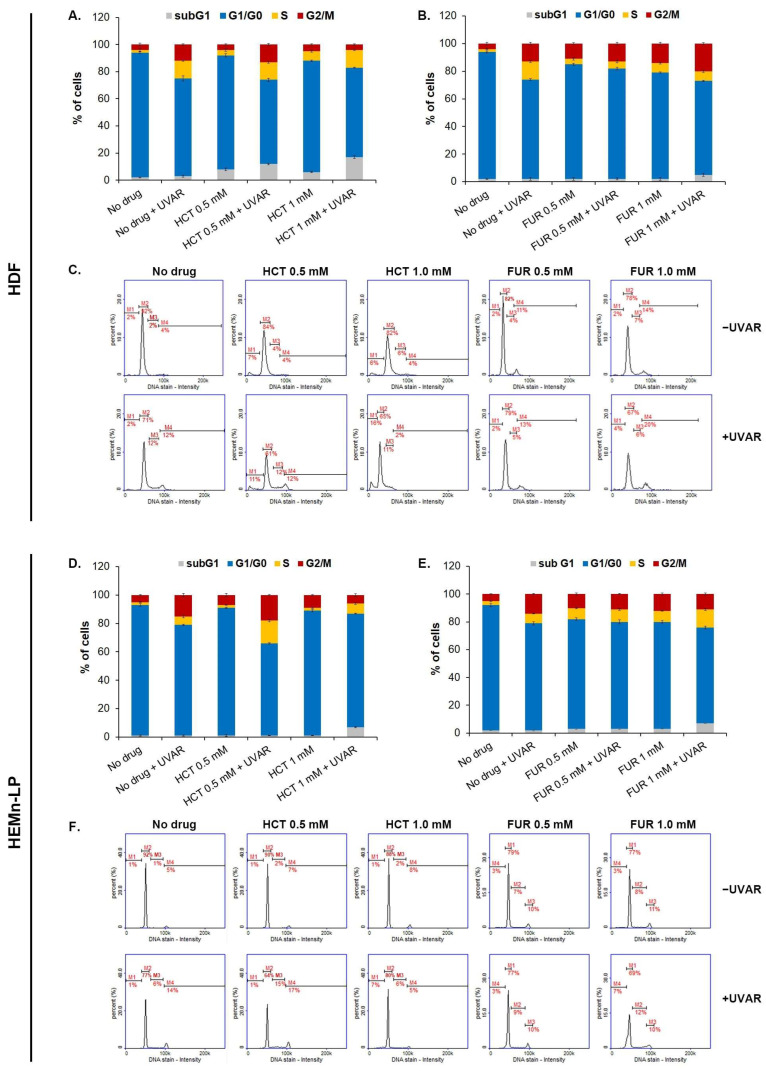
The influence of hydrochlorothiazide, furosemide as well as UVA irradiation on the cell cycle profile of melanocytes (**D**,**E**) and fibroblasts (**A**,**B**). Representative histograms depicting the distribution of cells in different phases of the cell cycle based on the different DNA content (**C**,**F**).

**Figure 5 ijms-25-01432-f005:**
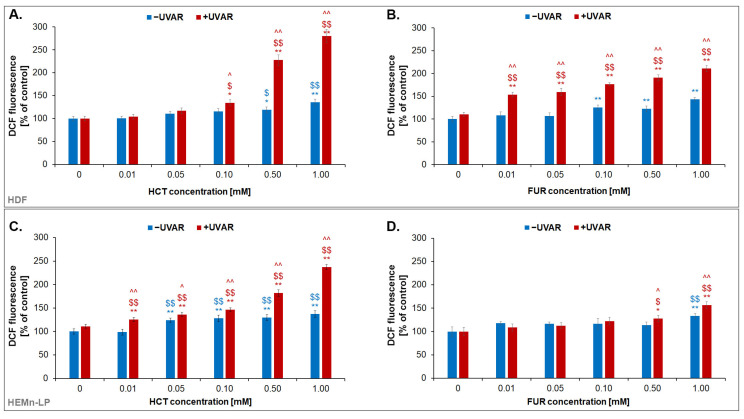
Quantitative analysis of the level of ROS in fibroblasts (**A**,**B**) and melanocytes (**C**,**D**) exposed to hydrochlorothiazide (0.05 mM and 1.0 mM) or furosemide (0.05 mM and 1.0 mM) as well as UVA irradiation using H_2_DCFDA probe. * *p* < 0.05, ** *p* < 0.01 vs. untreated cells (control); $ *p* < 0.05, $$ *p* < 0.01 vs. irradiated cells (non-treated cells); ^ *p* < 0.05, ^^ *p* < 0.01 vs. corresponding sample not treated with UVAR.

**Figure 6 ijms-25-01432-f006:**
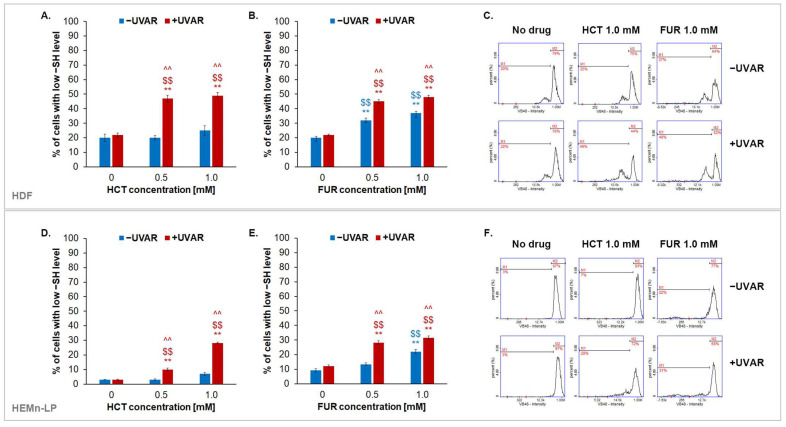
The evaluation of intracellular reduced thiol levels in fibroblasts (**A**,**B**) and melanocytes (**D**,**E**) incubated with hydrochlorothiazide, furosemide, and/or exposed to UVA radiation. Representative histograms differentiate cells with low and high content of reduced thiols based on Vita-Bright 48™ intensity (**C**,**F**). ** *p* < 0.01 vs. untreated cells (control); $$ *p* < 0.01 vs. irradiated cells (non-treated cells); ^^ *p* < 0.01 vs. corresponding sample not treated with UVAR.

**Table 1 ijms-25-01432-t001:** The effect of hydrochlorothiazide, furosemide, as well as UVA radiation on the viability of human fibroblasts and melanocytes.

	Relative Cell Viability (% of Control)	
	HDF	HEMn-LP
No drug	100.0	100.0
No drug + UVAR	99.9	101.4
HCT 0.5 mM	99.7	101.7
HCT 0.5 mM + UVAR	99.6	102.5
HCT 1 mM	91.8	98.4
HCT 1 mM + UVAR	58.6	90.5
FUR 0.5 mM	100.5	96.0
FUR 0.5 mM + UVAR	98.9	88.5
FUR 1 mM	98.0	94.3
FUR 1 mM + UVAR	104.5	82.7

Abbreviations: UVAR—UVA radiation; HCT—hydrochlorothiazide; FUR—furosemide; HDF—human dermal fibroblasts; HEMn-LP—human epidermal melanocytes lightly pigmented.

## Data Availability

The data that support the findings of this study are available from the corresponding author upon reasonable request.

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
