# Peer review of "Phototoxic Reactions Inducted by Hydrochlorothiazide and Furosemide in Normal Skin Cells—In Vitro Studies on Melanocytes and Fibroblasts"

_ijms, 2024, doi:10.3390/ijms25031432_

Round 1
Reviewer 1 Report
Comments and Suggestions for Authors
The authors assessed the phototoxic potential of two diuretics, furosemde and hydrochlorothiazide, using various parameters on cultured human fibroblasts and melanocytes.
Major comments :
· The phototoxicity of these drugs is well known, and is mediated by an oxidative stress. So, this study doesn't improve our knowledge on the adverse reactions of these drugs. Regarding the lower phototoxic potential of furosemide, this was also anticipated due to the dual phototoxic/antioxidant properties of furosemide. In the discussion, which is too long and not focused on the data, the authors should highlight the true novelty of their data, instead of reviewing the hypertension and known properties of these drugs.
Minor comments :
· Method 4.4, analysis of cell number, lines 446-447 : DAPI is actually a cell permeable dye, in particular depending on the conditions of its application. So, it is not easy to use it as an index of non-viable cells. In this case, it is important to describe the precise conditions of its application to ensure a good discrimination between viable and non-viable cells. This is important to analyse the Figure 2.
· Method 4.8, ROS quantitation, line 483 : H2DCFDA is actually the reduced form of 2',7'-dichlorofluorescein diacetate (DCFDA), and not that of fluorescein.
· Results, Figure 1A-1D, and Table 1 : in the absence of drug and UV exposure - which should correspond to the control - the % of control is not 100%; so, how was the control condition defined ?
Author Response
Dear Reviewer,
We would like to thank the Reviewer for the assessment of our manuscript. We are grateful for the comments and advices. According to the suggestions, we introduced following changes and corrections to the paper. All edited fragments of the manuscript are to be founds in bold.
COMMENT: The phototoxicity of these drugs is well known, and is mediated by an oxidative stress. So, this study doesn't improve our knowledge on the adverse reactions of these drugs. Regarding the lower phototoxic potential of furosemide, this was also anticipated due to the dual phototoxic/antioxidant properties of furosemide. In the discussion, which is too long and not focused on the data, the authors should highlight the true novelty of their data, instead of reviewing the hypertension and known properties of these drugs.
RESPONSE: The Authors have changed significant parts of the discussion in accordance with the Reviewer's recommendations. All changes made in the discussion section are marked in bold. We hope that in the current version the discussion better highlights the novelty of the presented research and focuses on the results obtained.
Drug-induced photosensitivity which is a dermatological growing problem is one of the most common cutaneous ADRs type [8]. The reasons for the increasing occurrence of phototoxicity cases include an increased content of substances with phototoxic potential in food and cosmetic products and more frequent exposure to UV radiation caused by the preference for dark skin tone. Also, knowledge about the common deficiency of vitamin D3 makes society more often expose its skin to UV radiation in order to equalize the its level. In addition, there are many over-the-counter drugs from various therapeutic groups available in the pharmaceutical market, which increase the percentage of society using pharmacotherapy without appropriate medical care. Therefore, occurring cutaneous ADRs contribute to patients discontinuing pharmacotherapy or using additional drugs aimed at alleviating the accompanying ADRs, directly affecting its effectiveness [19,20].
Hypertension is known to be a growing health problem worldwide. It is a complex disease associated with multiple abnormalities in metabolic, hemodynamic, and neuroendocrine systems leading to the development of cardiovascular diseases [21,22]. The first-line drugs recommended for the treatment of hypertension both in United Stated and Europe include diuretics like hydrochlorothiazide (HCT) and furosemide (FUR) [14,21,23,24,25]. Due to the fact that multi-morbidity associated with polypharmacy mainly affects the elderly population with an increased risk of heart disease, cutaneous side effects mainly concern this social group. It is worth noting that effective treatment of hypertension involves the use of several drugs such as beta-blockers, diuretics and calcium channel blockers, most of which have proven phototoxic properties. Exposure of the skin to many factors inducing phototoxic reactions contributes to the skin's sensitivity to the toxic effects of UV radiation, predisposing it to the occurrence of phototoxic reactions and the induction of damage to the genetic material that may result in the development of cancer [26,27].
Although both commonly known and used drugs such as HCT and FUR are initially subjected to phototoxicity test, many new reports, systematic reviews, and cohort studies of phototoxic reactions have been reported as an adverse effects of pharmacotherapy with these drugs [14,27,28,29]. Moreover, a lot of recent pharmacoepidemiological data have appeared to link a dose-dependent, cumulative association between long-therm HCT treatment and increased risk of skin cancer [24,27,28,29]. The meta-analyses of the relationship between pharmacotherapy with thiazide diuretics and the risk of skin cancer demonstrated an increased risk of developing non-melanoma skin cancers - squamous cell carcinoma (SCC) and basal cell carcinoma (BCC), for which the OR value was 1.86 and 1.19, respectively, and melanoma cancer (OR 1.14) [30]. This study confirmed the results of the presented meta-analyses at the molecular level by demonstrating greater resistance of melanocytes to the phototoxic effect of diuretics compared to fibroblasts. Based on the available WHO data, special precautions and warnings for the use of medications containing HCT have been added to the Summary of Product Characteristics in accordance with the European Medicine Agency Pharmacovigilance Risk Assessment Committee recommendations [24,31-34].
Despite the fact that HCT and FUR are drugs used mainly in the treatment of hypertension, renal dysfunction and edema, their phototoxic ADR is widely known [35]. However, currently available data do not indicate a direct effect of these drugs on normal skin cells. In this study, the cytotoxic and phototoxic potential of HCT and FUR against human normal skin cells with different content of melanin pigments was analysed for the first time. It was found that simultaneous exposure of human normal skin cells causes disturbances of intracellular redox homeostasis which was confirmed by quantifying the ROS levels and the assessment of reduced intracellular thiols. Furthermore, differences in cell cycle and mitochondrial potential induced by irradiated HCT and FUR in normal skin cells with different degrees of pigmentation were assessed. On the one hand, the photoprotective properties of melanins are related to their perinuclear localization, aimed at protecting DNA, but on the other hand, melanin has antioxidant properties, protecting the cell against the harmful effects of ROS [17,36,37]. Moreover, melanin biopolymers are able to bind and retain in the skin many substances, including drugs, creating long-term reservoir [38]. The formation of drug-melanin complexes may result in much higher concentrations of the drug in the skin, affecting the effectiveness and safety of the treatment, and increasing probability of phototoxic reactions occurrence. Drugs with confirmed ability to bind to melanins are NSAIDs, tetracyclines, and fluoroquinolones [39-41]. The observed changes demonstrated in this study suggest a significant contribution of melanin to the cellular response to HCT, FUR, and UVAR. It seems that melanin has photoprotective properties against the tested drugs, as fibroblasts turned out to be more sensitive cells to the tested agents, which is confirmed by the results of the studies discussed below and other tests conducted in vivo [42-44]. A similar experimental model was previously used to investigate the phototoxicity of fluoroquinolones and meloxicam against human normal skin cells. It has been shown that the tested drugs used in combination with UVA inhibit the proliferation of fibroblasts and melanocytes, and the obtained effect depends on the concentration of the analysed substances [45,46]. The presented results showing phototoxic properties of HCT and FUR was obtained during various analysis, including the number and viability of cells as well as microscopic images.
It is worth noting that other studies have proven that drugs containing a sulphonamide moiety, such as hydrochlorothiazide, cause skin reactions during the simultaneous irradiation with UVA. The changes within the skin induced by the phototoxicity are visible both macroscopically and histologically. Fibroblast morphology disorders including changes in cells shape, loss of intracellular connections and cell number reduction may indicate a significant participation of this group of cells in skin damage caused by HCT. Moreover, an in vitro assay analyzing the phototoxic potential of HCT indicated that the drug with UVAR caused dysplastic morphological changes in the keratinocytes HaCaT cell line [47].
Based on the obtained data, it was found that HCT, and FUR caused a reduction in the number of melanocytes and fibroblasts. The simultaneous irradiation with UVA contributed to the enhancing the effect. The observed decrease in the cell number may be the result of the inhibition of cell cycle. In all analyzes, the tested drugs reduced the percentage of cells in the G1/G0 phase, and UVA radiation additionally lowered the number of cells in this phase. In addition, the phototoxic action of HCT and FUR led to genetic material damage and DNA fragmentation as evidenced by an increase in the percentage of cells in the subG1 phase. Similar changes in the cell cycle were caused by phototoxic action of meloxicam in melanocytes and fibroblasts - the drug caused a decrease in the percentage of cells in the G1/G0 phase [45]. The analysis of the effect of lomefloxacin and UVAR on the fibroblast cell cycle showed similar results relating to the percentage of cells in the G1/G0 and G2/M phases as the HCT and FUR studies. Conversely, a decrease in the percentage of fibroblasts in the S phase was observed [48]. Although cell cycle disruption is one of the symptoms of phototoxicity, the direction of change may depend on the tested drug and used cell line.
Currently, two basic types of phototoxic reaction are distinguished - type I, oxygen dependent, (photodynamic), and type II, which does not depend on oxygen, (non-photodynamic) [8,49]. The first type of phototoxic reactions is accompanied by the generation of a large amount of ROS, which include hydroxyl peroxide, superoxide anion, as well as hydroxyl radical. The second type, non-photodynamic reaction, manifests in the generation of highly toxic and reactive singlet oxygen [50-52]. Both analysed diuretics are known to be photolabile compounds under aerobic and anaerobic conditions. During the irradiation, they are converted to unstable photoproducts, which contributes to the induction of phototoxicity reactions [47,50]. This study proved that HCT and FUR triggered generation of ROS in tested cultures, but their potential was different. HCT itself, in a dose-dependent manner, induced the production of ROS in both analysed cell lines, while FUR exhibited such properties only in relation to fibroblasts. In addition, HCT demonstrated a significantly greater pro-oxidative effect than FUR in irradiated cells. Significant differences in the results of H2DCFDA assay were noticed when compared cells treated only with the drug to cells exposed to the drug and UVAR. The results clearly indicated that UVA radiation is a factor enhancing the generation of ROS in treated cells. In addition, it was shown that the intracellular redox homeostasis of normal skin cells was disturbed, as evidenced by the high level of oxidized thiols. An imbalance between the amount of ROS generated and the cell ability to neutralize them is referred to as oxidative stress. The obtained results allow to conclude that HCT is the drug causing greater oxidation-reduction disorders than furosemide, while fibroblasts are more sensitive cells to oxygen homeostasis disorder than melanocytes. Many drugs with high phototoxic potential, i.e. lomefloxacin or meloxicam, induced disturbances of redox homeostasis in skin cells which resulted in an increase in ROS levels, a decrease in the percentage of cells with high levels of reduced thiols, similar to the tested HCT and FUR [45,48].
Considering the multitude of processes taking place with the participation of ATP-producing organelles, mitochondria are known to be the primary source of intracellular ROS [53,54]. ROS generated by mitochondria may regulate cell proliferation, among others by the induction of cell cycle arrest. Reduced polarization of mitochondrial membranes is observed in unfavorable conditions and is contributed by, among others, to induce oxidative stress and apoptosis [55]. The results presented in this article demonstrated that the tested drugs exhibited different potential on the influence on TMP. HCT turned out to reduce the mitochondrial membrane potential to a much greater extent than FUR. FUR lowered the mitochondrial membrane potential only in fibroblasts during the simultaneous UVA irradiation. The obtained results allow to conclude that one of the phototoxic effects of analysed diuretics on human normal skin cells is mitochondrial membrane depolarization.
In this study, for the first time, the cyto- and phototoxic potential of hydrochlorothiazide and furosemide was analysed on two human normal skin cell lines – melanocytes and fibroblasts, differentiated in terms of the content of melanin pigments. The obtained results showed that both drugs have antiproliferative potential against melanocytes and fibroblasts, and UVA radiation additionally contributes to its deepening. These results are reflected in the decrease in cell number and proliferation as well as the observed morphological changes. Simultaneous exposure of skin cells to the tested drugs and UVAR resulted in changes in the percentage of cells in individual phases of the cell cycle, a decrease in the transmembrane mitochondrial potential and changes in intracellular redox homeostasis. The tested drugs and UVAR significantly increase the level of intracellular ROS and reduce the percentage of cells with high levels of reduced thiols. The presented results indicate the phototoxic properties of the tested drugs, however, hydrochlorothiazide is a drug with a greater phototoxic potential and fibroblasts are more sensitive cells to the tested factors than melanocytes. We believe that these data provide a strong basis for further research on the phototoxic effects of furosemide and hydrochlorothiazide.
COMMENT: Method 4.4, analysis of cell number, lines 446-447: DAPI is actually a cell permeable dye, in particular depending on the conditions of its application. So, it is not easy to use it as an index of non-viable cells. In this case, it is important to describe the precise conditions of its application to ensure a good discrimination between viable and non-viable cells. This is important to analyse the Figure 2.
RESPONSE: The number of cells and their viability were determined using the image cytometer NucleoCounter® NC-3000™ in accordance with the manufacturer's protocol. Via1-Casettes™ filled with DAPI and acridine orange dyes were used for the assay. In general, the method is calibrated in a way that allows for precise assessment at the same time the number dead cells as well as the total cell number. Immediate measurement after loading the cassettes with cells makes that only dead cells with damaged cell membranes are stained with DAPI. Penetration of DAPI through cell membranes of live cells requires longer time. The method was developed and validated by Chemometec and is widely used to analyze cell viability and has also been used in previous studies.
The detailed analysis protocol for Cell Viability and Cell Count Assay is provided by the manufacturer in the link below:
https://chemometec.com/wp-content/uploads/2022/08/App-note_994-3011_Viability-and-Cell-Count-Via1-insect_NC-3000.pdf
COMMENT: Method 4.8, ROS quantitation, line 483: H2DCFDA is actually the reduced form of
2',7'-dichlorofluorescein diacetate (DCFDA), and not that of fluorescein.
RESPONSE: We would like to thank the Reviewer for pointing out the error in the methodology.
The correct version is below.
4.8 H2DCFDA assay – reactive oxygen species quantitation
The H2DCFDA fluorescent reagent was used to quantify intracellular ROS content. The analysis was carried out in accordance with the procedure described in an earlier article [33]. The assay involves staining cells with
2',7'-dichlorodihydrofluorescein diacetate (H2DCFDA), which is a reduced form of 2',7'-dichlorofluorescein diacetate (DCFDA). H2DCFDA is deacetylated by intracellular esterases to H2DCF and then oxidized to
2',7'-dichlorofluorescein (DCF), which exhibits green fluorescence. Cells after the HCT and FUR treatment and/or UVAR exposure were incubated with the H2DCFDA reagent for 30 min without light access. Subsequently, the cells were washed twice with PBS and fluorescence intensity was measured with Infinite 200 Pro microplate reader. The obtained re-sults were expressed as a percentage of control cells.
COMMENT: Results, Figure 1A-1D, and Table 1: in the absence of drug and UV exposure - which should correspond to the control - the % of control is not 100%; so, how was the control condition defined ?
RESPONSE: The improved version of Figure 1 A-D is included below. The results of the analysis of the metabolic activity of cells carried out using the WST-1 probe are calculated as a percentage of the control. The WST-1 assay is a preliminary, screening test aimed at assessing the general cyto- or phototoxic potential of a drug in a wide range of concentrations. The obtain results of the absorbance measurement were normalized to the percentage of control. In turn, the data presented in Table 1 show the cell viability as the direct results of the cell population analysis using image cytometer.
Figure 1. The evaluation of cytotoxic and phototoxic effects of hydrochlorothiazide and furosemide. Metabolic activity of skin cells - fibroblasts (A,B) and melanocytes (C,D) was assessed on the basis of the WST-1 assay. Representative photographs showing morphological changes in normal skin cells after the investigated treatment (E). * p < 0.05, ** p < 0.01 vs. untreated cells (control); $ p < 0.05,
$$ p < 0.01 vs. UVA-irradiated cells (non-exposed to the drug); ^^ p < 0.01 vs. corresponding sample not irradiated with UVAR.
Reviewer 2 Report
Comments and Suggestions for Authors
Hydrochlorothiazide (HCT) and furosemide (FUR) are both diuretic medications commonly used to treat conditions such as hypertension and edema. While these medications primarily affect the renal system, there is limited information on their direct cytotoxic effects on skin cells.
There are some minor comments for improvement of this manuscript.
- Including specific quantitative data or statistical measures (if available) in the abstract could enhance the completeness of the information.
- Establish a stronger link between hypertension and the phototoxic effects of HCT and FUR. Discuss the relevance of these findings to individuals with hypertension and the potential impact on their overall health.
- Expand on the discussion regarding drug-induced photosensitivity. Provide additional context on why this is a growing dermatological concern and how it impacts patient management and treatment decisions.
Author Response
Dear Reviewer,
The Authors would like to thank the Reviewer for all insightful suggestions and comments.
We have revised the manuscript following the Reviewer’s remarks.
COMMENT: Including specific quantitative data or statistical measures (if available) in the abstract could enhance the completeness of the information.
RESPONSE: The Authors supplemented the abstract with additional information and numerical values in accordance with the Reviewer's comments. The current version of the abstract with changes marked in bold is included below.
Abstract: Hypertension is known to be a multifactorial disease associated with abnormalities in neuroendocrine, metabolic, and hemodynamic systems. Poorly controlled hypertension causes more than one in eight premature deaths worldwide. Hydrochlorothiazide (HCT) and furosemide (FUR), being first-line drugs in the treatment of hypertension, are among others the most frequently prescribed drugs in the world. Currently, many pharmacoepidemiological data associate the use of these diuretics with an increased risk of adverse phototoxic reactions that may induce the development of melanoma and non-melanoma skin cancers. In this study, the cytotoxic and phototoxic potential of HCT and FUR against skin cells varied by melanin pigment content was assessed for the first time. The results showed that both drugs reduced the number of metabolically active normal skin cells in a dose-dependent manner. UVA irradiation significantly increased the cytotoxicity of HCT towards fibroblasts by approximately 40% and melanocytes by almost 20% compared to unirradiated cells. In the case of skin cells exposed to FUR and UVA radiation, a decrease in the number of metabolically active fibroblasts by approximately 35% and melanocytes by almost 20% was observed. Simultaneous exposure of melanocytes and fibroblasts to HCT or FUR and UVAR caused a decrease in cell viability, and number, which was confirmed by microscopic assessment of morphology. The phototoxic effect of HCT and FUR was associated with the disturbance of redox homeostasis confirm the oxidative stress as a mechanism of phototoxic reaction. UVA-irradiated drugs induced the generation of ROS by 50-200%, and oxidize intracellular thiols. A reduction in mitochondrial potential of almost 80% in melanocytes exposed to HCT and UVAR and 60% in fibroblasts was found due to oxidative stress occurrence. In addition, HCT and FUR have been shown to disrupt the cell cycle of normal skin cells. Finally, it can be concluded that HCT is the drug with a stronger phototoxic effect, and fibroblasts turn out to be more sensitive cells to the phototoxic effect of tested drugs.
COMMENT: Establish a stronger link between hypertension and the phototoxic effects of HCT and FUR. Discuss the relevance of these findings to individuals with hypertension and the potential impact on their overall health.
RESPONSE: We would like to thank the Reviewer for the tips that allowed us to improve the quality of the discussion. We hope that the added fragments indicate a relationship between the use of HCT and FUR in the treatment of hypertension and the increased risk of phototoxic reactions, which may induce cancerogenesis.
[…]Because multi-morbidity associated with polypharmacy mainly affects the elderly population with an increased risk of heart disease, cutaneous side effects mainly concern this social group. It is worth noting that effective treatment of hypertension involves the use of several drugs such as beta-blockers, diuretics, and calcium channel blockers, most of which have proven phototoxic properties. Exposure of the skin to many factors inducing phototoxic reactions contributes to the skin's sensitivity to the toxic effects of UV radiation, predisposing it to the occurrence of phototoxic reactions and the induction of damage to the genetic material that may result in the development of cancer [26,27]. […]
[…] The meta-analyses of the relationship between pharmacotherapy with thiazide diuretics and the risk of skin cancer demonstrated an increased risk of developing non-melanoma skin cancers - squamous cell carcinoma (SCC) and basal cell carcinoma (BCC), for which the OR value was 1.86 and 1.19, respectively, and melanoma cancer (OR 1.14) [30]. This study confirmed the results of the presented meta-analyses at the molecular level by demonstrating greater resistance of melanocytes to the phototoxic effect of diuretics compared to fibroblasts. […]
COMMENT: Expand on the discussion regarding drug-induced photosensitivity. Provide additional context on why this is a growing dermatological concern and how it impacts patient management and treatment decisions.
RESPONSE: The Authors added information in the Discussion section about the reasons for the increased incidence of phototoxicity and the impact of occurring side effects on the health status and effectiveness of pharmacotherapy of patients. The completed fragments are marked in bold.
[…] The reasons for the increasing occurrence of phototoxicity cases include an increased content of substances with phototoxic potential in food and cosmetic products and more frequent exposure to UV radiation caused by the preference for dark skin tone. Also, knowledge about the common deficiency of vitamin D3 makes society more often expose its skin to UV radiation in order to equalize its level. In addition, there are many over-the-counter drugs from various therapeutic groups available in the pharmaceutical market, which increase the percentage of society using pharmacotherapy without appropriate medical care. Therefore, occurring cutaneous ADRs contribute to patients discontinuing pharmacotherapy or using additional drugs aimed at alleviating the accompanying ADRs, directly affecting their effectiveness [19,20]. […]
References
[19] Drucker, A.M.; Rosen, C.F. Drug-induced photosensitivity: Culprit drugs, management and prevention. Drug Saf 2011, 34, 821–837.
[20] Moore, D.E.; Drug-induced cutaneous photosensitivity: incidence, mechanism, prevention and management. Drug Saf 2002;25(5):345–72.
[26] Lugović-Mihić, L.; Duvančić, T.; Ferček, I.; Vuković, P.; Japundžić, I.; Ćesić, D. Drug-induced photosensitivity - a continuing diagnostic challenge. Acta Clin Croat 2017, 56(2), 277-283.
[27] Kaae, J.; Boyd, H.A.; Hansen, A,V.; Wulf, H.C.; Wohlfahrt, J.; Melbye, M. Photosensitizing medication use and risk of skin cancer. Cancer Epidemiol Biomarkers Prev 2010, 11, 2942-2949.
[30] Shin, D.; Lee, E.S.; Kim, J.; Guerra, L.; Naik, D.; Prida, X. Association between the use of thiazide diuretics and the risk of skin cancers: a meta-analysis of observational studies. J Clin Med Res 2019, 11, 247–255.
Round 2
Reviewer 1 Report
Comments and Suggestions for Authors
· Abstract : In the new sentence in the Abstract, the description of UVA exposure for HCT and FUR are different, which makes the sentence somewhat confusing. It should be the same description for both. Moreover, I don't understand how the differences between controls were calculated, which do not correspond to the data presented in Table 1. I still don't understand why the no UV-no drug controls are not 100%.
· Abstract, lines 27-28, ROS description : Replace "induced" by "increased"(because control is not zero); according to the Figure 5, the increase of ROS is between 10% and 150%, depending on the cells and the compound (and not 50-200%).
· Figure 5 : As discussed previously, the Y-axis does not indicate a unit of ROS, but variations of fluorescence units; so, Y-axis should mention "DCF fluorescence [% of control]; the title of the graph may be "ROS generation".
Author Response
Dear Reviewer,
We introduced the following changes to the manuscript in line with the Reviewer's suggestions.
COMMENT: Abstract: In the new sentence in the Abstract, the description of UVA exposure for HCT and FUR are different, which makes the sentence somewhat confusing. It should be the same description for both.
Abstract, lines 27-28, ROS description : Replace "induced" by "increased"(because control is not zero); according to the Figure 5, the increase of ROS is between 10% and 150%, depending on the cells and the compound (and not 50-200%).
RESPONSE: The Authors assure that all suggestions proposed by the Reviewer in the Abstract have been introduced and marked in bold.
Abstract: Hypertension is known to be a multifactorial disease associated with abnormalities in neuroendocrine, metabolic, and hemodynamic systems. Poorly controlled hypertension causes more than one in eight premature deaths worldwide. Hydrochlorothiazide (HCT) and furosemide (FUR), being first-line drugs in the treatment of hypertension, are among others the most frequently prescribed drugs in the world. Currently, many pharmacoepidemiological data associate the use of these diuretics with an increased risk of adverse phototoxic reactions that may induce the development of melanoma and non-melanoma skin cancers. In this study, the cytotoxic and phototoxic potential of HCT and FUR against skin cells varied by melanin pigment content was assessed for the first time. The results showed that both drugs reduced the number of metabolically active normal skin cells in a dose-dependent manner. UVA irradiation significantly increased the cytotoxicity of HCT towards fibroblasts by approximately 40% and melanocytes by almost 20% compared to unirradiated cells. In the case of skin cells exposed to FUR and UVA radiation, an increase in cytotoxicity by approximately 30% for fibroblasts and 10% for melanocytes was observed. Simultaneous exposure of melanocytes and fibroblasts to HCT or FUR and UVAR caused a decrease in cell viability, and number, which was confirmed by microscopic assessment of morphology. The phototoxic effect of HCT and FUR was associated with the disturbance of redox homeostasis confirming the oxidative stress as a mechanism of phototoxic reaction. UVA-irradiated drugs increased the generation of ROS by 10-150%, and oxidized intracellular thiols. A reduction in mitochondrial potential of almost 80% in melanocytes exposed to HCT and UVAR and 60% in fibroblasts was found due to oxidative stress occurrence. In addition, HCT and FUR have been shown to disrupt the cell cycle of normal skin cells. Finally, it can be concluded that HCT is the drug with a stronger phototoxic effect, and fibroblasts turn out to be more sensitive cells to the phototoxic effect of tested drugs.
COMMENT: Moreover, I don't understand how the differences between controls were calculated, which do not correspond to the data presented in Table 1. I still don't understand why the no UV-no drug controls are not 100%.
RESPONSE: For greater clarity, the results presented in Table 1 were recalculated into % of control, (a value of 100% is for the sample not exposed to drugs nor UVA radiation). The Authors would like to emphasize that there is no direct connection between the results presented in Figure 1 and Table 1. The data presented come from two different experiments:
· Figure 1 presents results obtained by the spectrophotometric determining the number of metabolically active cells (screening assay made by the use of WST-1 reagent, which is converted by mitochondrial dehydrogenases)
·Table 1 presents the results of the cytometric analysis of cell viability (the percentage of live cells in the examined cell population). The assessment of cell viability is possible by determining the total cell count (cells stained with acridine orange) and the number of dead cells (cells stained with DAPI).
Table 1. The effect of hydrochlorothiazide, furosemide, as well as UVA radiation on the viability of human fibroblasts and melanocytes.
|
Relative cell viability |
|
|
HDF |
HEMn-LP |
No drug |
100.0 |
100.0 |
No drug + UVAR |
99.9 |
101.4 |
HCT 0.5 mM |
99.7 |
101.7 |
HCT 0.5 mM + UVAR |
99.6 |
102.5 |
HCT 1 mM |
91.8 |
98.4 |
HCT 1 mM + UVAR |
58.6 |
90.5 |
FUR 0.5 mM |
100.5 |
96.0 |
FUR 0.5 mM + UVAR |
98.9 |
88.5 |
FUR 1 mM |
98.0 |
94.3 |
FUR 1 mM + UVAR |
104.5 |
82.7 |
Abbreviations: UVAR – UVA radiation; HCT – hydrochlorothiazide; FUR – furosemide; HDF – human dermal fibroblasts; HEMn-LP – human epidermal melanocytes lightly pigmented |
COMMENT: Figure 5 : As discussed previously, the Y-axis does not indicate a unit of ROS, but variations of fluorescence units; so, Y-axis should mention "DCF fluorescence [% of control]; the title of the graph may be "ROS generation".
RESPONSE: The Authors changed the description of the Y axis in Figure 5 according to the reviewer's suggestion. The new version of Figure 5 is shown below.
Figure 5. Quantitative analysis of the level of ROS in fibroblasts (A,B) and melanocytes (C,D) exposed to hydrochlorothiazide (0.05 mM and 1.0 mM) or furosemide (0.05 mM and 1.0 mM) as well as UVA irradiation using H2DCFDA probe. * p < 0.05, ** p < 0.01vs. untreated cells (control); $ p < 0.05,
$$ p < 0.01 vs. irradiated cells (non-treated cells); ^ p < 0.05, ^^ p < 0.01 vs. corresponding sample not treated with UVAR.